# Mortality and Associated Factors in Patients with COVID-19: Cross-Sectional Study

**DOI:** 10.3390/vaccines11010071

**Published:** 2022-12-28

**Authors:** Vergílio Pereira Carvalho, João Paulo Jordão Pontes, Demócrito Ribeiro de Brito Neto, Celso Eduardo Rezende Borges, Gisele Ribeiro Londe Campos, Hugo Leonardo Shigenaga Ribeiro, Waldemar Naves do Amaral

**Affiliations:** 1Post-graduation course in Health Sciences at the Medical School, Federal University of Goiás (UFG), Goiânia 74690-900, GO, Brazil; 2Department of Anesthesiology, Training and Teaching Center in Anesthesiology, Uberlândia 38400-448, MG, Brazil; 3Department of Geriatrics, Brotherhood of the Santa Casa de Misericórdia de São Paulo, São Paulo 01221-010, SP, Brazil

**Keywords:** COVID-19, critical care, mortality, respiratory distress syndrome

## Abstract

The novel virus severe acute respiratory syndrome coronavirus 2 (SARS-COV-2) is highly virulent and causes coronavirus disease 2019 (COVID-19), resulting in high morbidity and mortality mainly associated with pulmonary complications. Because this virus is highly transmissible, it was quickly spread globally, resulting in COVID-19 being declared as a pandemic. This study aimed to analyze the prevalence of mortality and the factors related to mortality due to COVID-19 in patients with severe acute respiratory syndrome (SARS) at a university hospital in the Central—West region of Brazil. This retrospective cross-sectional study was based on an analysis of the medical records of patients with SARS aged >18 years and admitted to an intensive care unit due to COVID-19 with the requirement of invasive mechanical ventilation. Hospital death was considered as an outcome variable in this study. Moreover, demographic and lifestyle-related variables as well as the therapeutic measures used during the hospital stay were recorded and correlated with the death outcome. After excluding 188 medical records, 397 were analyzed. Most of the participants were men (59.7%), and the mortality rate in patients with SARS due to COVID-19 was 46.1%. Multiple regression analysis indicated that the independent factors associated with mortality in patients with SARS due to COVID-19 were the age of >60 years (*p* < 0.001) and the use of azithromycin (*p* = 0.012). Protective factors for mortality were considered as not having the following diseases: hyperthyroidism, asthma, hepatic inheritance, and not being a smoker. The mortality rate in patients with SARS due to COVID-19 was associated with older age and the use of azithromycin.

## 1. Introduction

The novel severe acute respiratory syndrome coronavirus 2 (SARS-CoV-2), which causes the infection called coronavirus disease 2019 (COVID-19) according to the World Health Organization, is one of the highly virulent β-coronaviruses that infects humans and causes acute respiratory failure (ARF) due to pneumonia, with high mortality rates [1,2].

The first case in Brazil was reported on 26 January 2020 and a public health emergency was declared on 3 February. In the 11th epidemiological week (SE), more than 100 cases were reported, and in the 12th SE, more than 1000 cases were reported. Community spread of COVID-19 was recognized in Brazil on March 20. In the 41st SE, Brazil ranked third in the world in notified cases (5.566.049) and second in number of deaths (160.496) [3].

An exact measurement of the number of deaths due to the COVID-19 pandemic is relevant for each country and region to understand the magnitude of the impact of the pandemic on public health. Accurate death estimation is also important for verifying the determinants of variation in the infection fatality rate across populations and has a direct contribution to predicting the pandemic and investigating alternative policy options. Reported deaths attempt to quantify the magnitude of the COVID-19 pandemic in different populations and locations over time, and they are widely seen as a more reliable indicator for tracking the pandemic relative to the reported case rates. However, reported deaths represent only a partial count of the total death toll from the COVID-19 pandemic, and the reliability of reported deaths varies greatly between locations and over time [4].

Excess mortality relates to the total number of deaths that occur during an interval above what is expected for that period. In a pandemic, it is known that the number of deaths tends to increase not only because of the disease, but also due to complications, such as deaths due to other comorbidities that were not adequately treated due to the person’s lifestyle, which is associated with the overloaded healthcare system. Thus, excess mortality is a relevant indicator of the social and economic impacts of a pandemic [5].

In addition, SARS-CoV-2 presents a major challenge for healthcare systems across the world. This is mainly because critically ill patients with ARF have to be monitored in an intensive care unit (ICU) mainly under the support of invasive mechanical ventilation (IMV) [2,3,4]. In addition, it is understood that independent factors associated with in-hospital mortality include: advanced age, male gender, immunosuppression, kidney disease, chronic lung disease, cardiovascular disease, cerebrovascular disease, and diabetes [6].

About 5% of COVID-19 patients and 20% of those hospitalized exhibit severe symptoms that need intensive care. More than 75% of patients hospitalized with COVID-19 require supplemental oxygen. Treatment for individuals with COVID-19 is based on best practices for the supportive management of acute hypoxic respiratory failure. Among patients admitted to the intensive care unit, mortality reaches 40%. At least 120 SARS-CoV-2 vaccines are in development. Although vaccination has not yet eradicated the COVID-19, the main means of slowing the spread of the virus are face masks, social distancing, and hand hygiene [7]. 

Studies infer that patients with prolonged treatment with angiotensin-converting enzyme inhibitors, angiotensin II blockers, beta-blockers, statins, diuretics, antiplatelet agents, and anticoagulants before ICU admission was associated with higher mortality. This is because angiotensin-converting enzyme 2 is the primary receptor for SARS-CoV-2 entry into host cells [2].

The diagnosis of COVID-19 is based on a reverse transcriptase polymerase chain reaction test [8]. To date, the treatment of COVID-19 is undefined, and the therapies used are aimed at controlling clinical manifestations and providing ventilator support. Notably, some drugs with therapeutic potential are emerging in the medical field [9]. 

Although many researchers have investigated SARS-CoV-2, [1,2,3,4] it is still important to explore the factors associated with mortality in patients with COVID-19, including the influence of sociodemographic conditions and the presence of chronic diseases, lifestyle, and treatments. Therefore, this study aimed to analyze the prevalence and factors related to mortality in patients with severe acute respiratory syndrome (SARS) due to COVID-19 requiring IMV. These data are important for adopting appropriate behavior in the hospital environment.

## 2. Materials and Methods

This is a cross-sectional observational study carried out in an ICU in the Midwest region of Brazil, located in the municipality of Rio Verde in the state of Goiás. The study was approved by the local Research Ethics Committee on 26 February 2021 (protocol number: 4.563.056/2021; Certificate of Presentation for Ethical Appraisal (CAAE) number: 43454621.2.0000.5077). Data collection was performed from 12 March 2021 to 26 May 2022.

The inclusion criteria were age 18 years or older; diagnosis of COVID-19 via PCR testing for SARS-CoV-2 from nasopharyngeal swabs or computed tomography scan findings compatible with the disease (bilateral multifocal ground-glass opacities ≥50%); and diagnosis of flu syndrome with institutional criteria for hospitalization on hospital admission, presenting a respiratory rate greater than 24/min, saturation less than 93% while breathing room air, or risk factors for complications (e.g., heart disease, diabetes, systemic arterial hypertension, neoplasms, immunosuppression, pulmonary tuberculosis, and obesity) followed by COVID-19 confirmation, and only patients admitted to the ICU. Patients were excluded if they who had three or more incompletely analyzed variables in their medical records, who had no record of death outcome due to transfer to another institution, or who demonstrated an unclear association between death and SARS-CoV-2 infection.

Patients who had clinical dyspnea and required supplemental oxygen and and the ratio of partial arterial pressure of oxygen (PaO_2_) and fraction of inspired oxygen (FiO_2_) ≤ 300 mmHg were considered as having ARF [10]. 

Data regarding sociodemographic characteristics, chronic diseases, lifestyle, treatment, and hospital death were collected from medical records. In-hospital death was considered as an outcome or a dependent variable. To provide a demographic characterization, the variables sex (male or female) and age were used. The considered chronic disease variables were systemic arterial hypertension; diabetes mellitus (DM); hypothyroidism; hyperthyroidism; chronic obstructive pulmonary disease; cardiovascular disease (CVD); cerebrovascular disease; mental health-related disease; a history of thrombosis; hepatitis A, B, and C; asthma; and liver failure. The considered lifestyle variables were smoking, alcohol consumption, and illicit drug use. The considered treatment variables were the use of the prone position during hospitalization and the use of drugs, such as dexamethasone, azithromycin, chloroquine, orhydroxychloroquine, and low-molecular-weight heparin (enoxaparin).

For the sample size calculation, an expected prevalence (or rate) of death of 45% was used [8], with an error margin of 5.0% and a 95% confidence interval (95% CI) for a population of 300,000 inhabitants residing in the city of Rio Verde, State of Goiás, Brazil, and its neighboring districts. Thus, the sample size required for this analysis (prevalence calculation) was found to be 380 patients.

All statistical analyses were performed using the Stata statistical package, version 16.0 (StataCorp LLC, College Station, TX, USA). The variables were presented as absolute numbers (*n*) and relative frequencies (%) with mean and standard deviation. Poisson’s regression was used to calculate the prevalence ratio and 95% CI, and *p*-values was obtained using the Wald test. Variables with a *p*-value of <0.20 in bivariate analysis were included in multiple hierarchical Poisson regression analyses, with robust variance based on a hierarchical model [9]. The independent variables in this hierarchical analysis were classified as follows: (I) demographic data (gender and age), (II) chronic diseases (DM and mental illness), (III) clinical features (pronation), and (IV) medication use (azithromycin and heparin). The effect size was calculated for the chi-square goodness of fit and independence tests using the *Phi *φ.** The *p*-value of 0.001 indicates that the relationship is statistically significant at the α = 0.05 level. If the alpha error is 0.05, the *p*-value of 0.001 is incorrect or needs to be clarified. Thus, it is understood that the effect sizes either measure the sizes of associations between variables or the sizes of differences between group means. An effect size ≥ 0.10 is considered small, ≥ 0.30 medium, and ≥ 0.50 large.

Variables without statistical power, that is, variables that after the bivariate analysis presented *n* < 10 in any stratum, were excluded from the multiple regression analysis [10]. 

## 3. Results

Overall, 585 medical records were evaluated for eligibility in March 2021; of these, 188 were excluded (Figure 1). Thus, 397 patients admitted to the ICU with SARS due to COVID-19 and in need of IMV participated in this study. Of these, 59.70% were men with a mean age of 61.04 ± 0.76 years (range 20–111 years). Of the patients aged >60 years, 62.17% were men and 37.83% were women.

The prevalence of death in individuals with SARS due to COVID-19 was 46.10% (63.93% in men and 36.07% in women). Data regarding the prevalence and association of in-hospital death in patients with SARS due to COVID-19 are presented in Table 1. The bivariate analysis indicated that in-hospital death in patients with SARS due to COVID-19 was associated with the age of >60 years (prevalence ratio [PR] = 1.61; 95% CI: 1.26–2.04, *p* < 0.001). In addition, not having a diagnosis of hyperthyroidism (PR = 0.46; 95% CI: 0.41–0.51, *p* < 0.001), asthma (PR = 0.46; 95% CI: 0.41–0.51, *p* < 0.001), or liver failure (PR = 0.46; 95% CI: 0.41–0.51, *p* < 0.001) and not being a smoker (PR = 0.61; 95% CI: 0.46–0.82, *p* = 0.002) were considered protective factors for mortality in individuals with SARS due to COVID-19.

Pronation at some point during hospitalization (PR = 1.30; 95% CI: 1.03–1.64, *p* = 0.028), the use of azithromycin (RP = 1.36; 95% CI: 1.09–1.69, *p* = 0.005), and use of chloroquine or hydroxychloroquine (PR = 1.76; 95% CI: 1.121–2.77, *p* = 0.014) were associated with the mortality of patients hospitalized with SARS due to COVID-19.

The variables included in the multiple regression analysis are shown in Table 2. After the multiple regression analysis, the age of >60 years (*p* < 0.001) and the use of azithromycin (*p* = 0.012) remained independently associated with the mortality of patients with SARS due to COVID-19.

## 4. Discussion

The main results of the present study indicate a high prevalence of death in individuals with SARS due to COVID-19 who required IMV. The mortality rate was higher in individuals aged >60 years who received azithromycin during SARS-CoV-2 infection.

The prevalence of death (46.1%) in individuals hospitalized with SARS due to COVID-19 was similar to that reported in other studies [2,8]. Notably, the patients included in the present study were critically ill and treated with IMV on admission to the ICU. These data are concerning and underscore that the high mortality due to COVID-19 in critically ill patients may be because of the continuous need for respiratory support and long stays in an ICU [2]. 

One study showed the relationship between COVID-19 mortality rates, the number of hospital beds, the number of general practitioners, and, therefore, the organization of the regional health system. The Italian regions with the highest levels of centralized healthcare, represented by the number of hospital beds, experienced a higher number of deaths, while regions with greater community support, exemplified by the number of general practitioners, had higher survival. However, it is important to identify that in ICU care with doctors specialized in intensive care medicine, this contributes beneficially to the health care of these patients. This is in line with our study, carried out in a city in the interior of Goiás, where many patients were waiting for the release of an ICU bed [4].

The findings of the present study confirm that the survival of critically ill patients with COVID-19 is particularly low in elderly men. Moreover, age of >60 years was associated with in-hospital death. A systematic review also stated that advanced age has been recognized as an important risk factor for COVID-19 mortality [4]. One explanation is that aging leads to impaired functioning of multiple body systems, including the immune system, which is a factor involved in increased mortality due to COVID-19 in the elderly [11]. 

In November 2020, Brazil reached third place in the world in the number of cases of COVID-19 and second place in the number of deaths from the disease. A descriptive study was carried out on deaths, mortality rates, years of potential life lost (YYL), and excess mortality from COVID-19, based on SARS-CoV-2 records at SIVEP-Gripe (Ministério da Saúde do Brasil) from February 16, 2020 to January 1, 2021. During this period, there were 98,025 deaths from COVID-19 in Brazil. Men represented 60.5% of the estimated 1.2 million PYLLs. High PYLL means evidenced prematurity of deaths. The population aged 45 to 64 years (both sexes) represented more than 50% of the total PYLL. Risk factors were present in 69.5% of deaths, with heart disease, diabetes, and obesity being the most prevalent comorbidities in both sexes. Indigenous people had the lowest number of deaths and the highest mean PYLL. However, among indigenous people, pregnant women and mothers had an average PYLL of more than 35 years [3].

Although mortality was higher in men, no statistical significance was observed in the present study. Grasselli et al. found a relationship between the male sex and increased mortality in patients with COVID-19 admitted to ICUs [2]. In addition to higher mortality, another large study involving 3.1 millions of patients with COVID-19 reported that the male sex was associated with a higher ICU admission rate [11]. This phenomenon is attributed to the fact that women have a higher number of CD4+ T cells, a more robust cytotoxic activity of CD8+ T cells, and greater production of immunoglobulin by B cells than men, enabling them to produce a more efficient cellular and humoral response [12]. 

The use of azithromycin was also associated with in-hospital death. Azithromycin is an antibiotic with anti-inflammatory and antiviral properties, and it was thought to have activity against SARS-CoV-2 [13]. However, based on different studies, a systematic review reported that azithromycin, along with other medications, including angiotensin-converting enzyme inhibitors, aspirin, colchicine, hydroxychloroquine, inhaled corticosteroids, intranasal corticosteroids, interferon beta, ivermectin, lopinavir-ritonavir, and vitamin C, has no important benefit on any important outcome for patients with COVID-19 [14]. Based on the present study design, this association between azithromycin and death does not indicate causality.

Dexamethasone may be beneficial in patients with COVID-19, mainly in more severe forms with exacerbated inflammatory activity, because of its potential anti-inflammatory effect, which confers its ability to decrease the gene transcription of several proinflammatory cytokines, chemokines, and adhesion molecules by inhibiting the generation and release of these mediators. However, dexamethasone may also prevent B-cell-mediated antibody production and reduce T-cell immune function, which may result in a higher plasma viral load and an increased risk of secondary infections [15]. Although the present study did not report an association of dexamethasone use with mortality, a randomized clinical trial in patients hospitalized with COVID-19 in the United Kingdom comparing the use of 6 mg dexamethasone once daily for 10 days to a placebo reported reduced 28-day mortality rate in patients with COVID-19 receiving either IMV or oxygen therapy without IMV, with no impact on mild cases without respiratory support [16].

Based on the current evidence, low-dose systemic steroids may be considered for specific patients with COVID-19 who are critically ill or require supplemental oxygen. However, routine use of corticosteroids should be avoided, particularly in patients with mild symptoms or in the early stages of the disease unless indicated for another reason, such as those related to an individual’s condition [17,18]. The retrospective nature of this study and the lack of standardization of doses and durations of therapy may have contributed to the absence of any association of corticosteroid use with mortality in the present study. This is because any use of dexamethasone, whether during ICU stay or earlier, was considered in this study.

The prone position contributes positively to the ventilation-perfusion ratio and to the recruitment of dependent lung segments, culminating in the opening of collapsed dependent alveoli and thereby providing better gas exchange and oxygenation. Among mechanically ventilated non-COVID-19 patients with severe acute respiratory distress syndrome, those who were ventilated in the prone position had a lower mortality rate [6]. The prone position can reduce the relative fraction of the pulmonary shunt by 30% compared with the supine group in patients with injured lungs [19]. A recent meta-analysis demonstrated that the prone position improved the PaO₂/FiO₂ ratio, with better SpO₂ than the supine position, in patients with COVID-19 [20]. Despite the aforementioned beneficial effects, the present study found an association between pronation and death, which may actually reflect the greater severity in patients who were pronated to improve ventilation and gas exchange. However, after the multiple regression analysis, pronation was not confirmed as an independent factor associated with mortality.

The limitations of this study include the observational cross-sectional design, which made it impossible to establish a causal relationship between the variables. In addition, the data collection considered a sample from a medium-sized city in the State of Goiás; therefore, generalization of the results to the rest of the Brazilian population must be completed with caution. Variables that included previous lung disease had a low prevalence; therefore, it was impossible to establish an association with the outcome variable or to associate the use of drugs, such as chloroquine and hydroxychloroquine, with mortality. In addition, variables, such as the length of hospital stay, ventilation parameters, and laboratory tests and vaccination, were not examined. In addition, understanding mortality and its associated factors can be used to guide the prioritization of health interventions, such as prioritizing vaccination, blockades, or the distribution of family members, to those who need them the most. Inequalities and a lack of adequate healthcare resources, hospital beds, and personal protective equipment varied by region in Brazil. The politicization of COVID-19 and the lack of a coherent national pandemic plan is a factor to be taken into account. The authors recommend examining these variables in future studies.

## 5. Conclusions

The authors conclude that individuals with SARS due to COVID-19 requiring IMV have high mortality. In addition, an association of mortality with an age of >60 years and the use of azithromycin was observed. The protective factors for mortality were considered as not having the following diseases: hyperthyroidism, asthma, hepatic inheritance, and not being a smoker.

## Figures and Tables

**Figure 1 vaccines-11-00071-f001:**
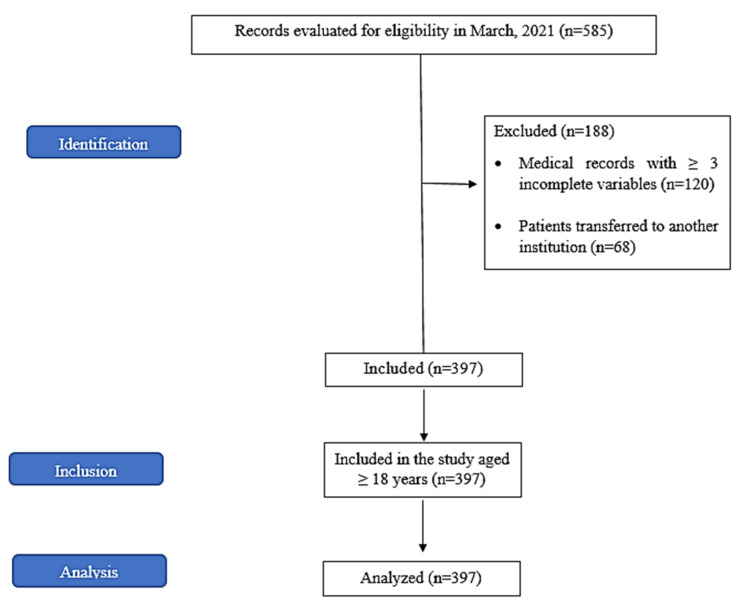
Study conduction flowchart according to the Strengthening the Reporting of Observational Studies in Epidemiology (STROBE) guidelines.

**Table 1 vaccines-11-00071-t001:** Association between hospital death in patients with severe acute respiratory syndrome owing to COVID-19 and sociodemographic data, chronic diseases, lifestyle, and treatment (*n* = 397).

		Hospital Death			
Variable	Frequencyn (%)	Prevalence *n* (%)	PR (CI 95%)	Effect Size (Phi φ)	*p*
Gender				0.124	0.119
Male	237 (59.70)	117 (63.93)	1.20 (0.95–1.50)		
Female	160 (40.30)	66 (36.07)	1		
Age				**0.200**	**0.001**
20–59	167 (42.07)	57 (31.15)	1		
60 or more	230 (57.93)	126 (68.85)	1.61 (1.26–2.04)		
Systemic arterial hypertension				0.083	0.228
No	184 (46.46)	91 (49.45)	1		
Yes	212 (53.54)	92 (43.40)	1.139 (0.85–1.52)		
Diabetes mellitus				0.145	0.082
No	275 (69.27)	64 (34.97)	1		
Yes	122 (30.73)	119 (65.03)	1.21 (0.98–1.50)		
Hypothyroidism				0.040	0.800
No	374 (94.21)	173 (94.54)	1.06 (0.66–1.72)		
Yes	23 (5.79)	10 (5.46)	1		
Hyperthyroidism				**0.060**	**0.001**
No	396 (99.75)	182 (99.45)	0.46 (0.41–0.51)		
Yes	1 (0.25)	1 (0.55)	1		
Chronic obstructive pulmonary disease				0.015	0.622
No	363 (91.44)	166 (90.71)	0.91 (0.64–1.30)		
Yes	34 (8.56)	17 (9.29)	1		
Cardiovascular disease				0.040	0.546
No	356 (89.67)	166 (90.71)	1.12 (0.77–1.64)		
Yes	41 (10.33)	17 (9.29)	1		
Cerebrovascular disease				0.042	0.595
No	377 (94.96)	175 (95.63)	1.16 (0.67–2.00)		
Yes	20 (5.04)	8 (4.37)	1		
Mental-health-related illness				0.065	0.185
No	384 (96.73)	175 (95.63)	0.74 (0.47–1.15)		
Yes	13 (3.27)	8 (4.37)	1		
Thrombosis history				-	-
No	396 (99.75)	183 (100.00)	-		
Yes	1 (0.25)	0 (0.00)	-		
Hepatitis A				0.020	0.469
No	392 (98.74)	180 (98.36)	1.31 (0.63–2.69)		
Yes	5 (1.26)	3 (1.64)	1		
Hepatitis B				0.020	0.469
No	392 (98.74	180 (98.36)	1.31 (0.63–2.70)		
Yes	5 (1.26)	3 (1.64)	1		
Hepatitis C				0.020	0.469
No	392 (98.74)	180 (98.36)	1.31 (0.63–2.70)		
Yes	5 (1.26)	3 (1.64)	1		
Asthma				**0.080**	**0.001**
No	391 (98.49)	179 (97.81)	0.46 (0.41–0.51)		
Yes (being treated)	1 (0.25)	1 (0.55)	1		
Yes (cured)	5 (1.26)	3 (1.64)	0.59 (0.29–1.23)		
Liver failure				**0.110**	**0.001**
No	390 (98.24)	178 (97.27)	0.46 (0.41–0.51)		
Yes (being treated)	1 (0.25)	1 (0.55)	1		
Yes (cured)	6 (1.51)	4 (2.19)	0.66 (0.38–1.17)		
Smoking				**0.398**	**0.002**
No	369 (92.95)	167 (91.26)	0.61 (0.46–0.82)		
Yes (smoker)	9 (2.27)	2 (1.09)	0.30 (0.09–1.06)		
Yes (used to smoke)	19 (4.79)	14 (7.65)	1		
Alcohol consumption				0.576	0.397
No	345 (86.90)	162 (88.52)	1.16 (0.82–1.65)		
Yes	52 (13.10)	21 (11.48)	1		
Use of illicit drugs				0.080	0.294
No	388 (97.73)	177 (96.72)	0.68 (0.30–1.53)		
Yes (user)	3 (0.76)	2 (1.09)	1		
Yes (used to use)	6 (1.51)	4 (2.19)	0.67 (0.30–1.48)		
Pronation				**0.101**	**0.028**
No	154 (38.99)	60 (32.97)	1		
Yes	241 (61.01)	122 (67.03)	1.30 (1.03–1.64)		
Dexamethasone use				0.039	0.861
No	79 (19.95)	37 (20.33)	1		
Yes	317 (80.05)	145 (79.67)	0.98 (0.75–1.27)		
Use of Azithromycin				**0.361**	**0.005**
No	207 (52.41)	81 (44.75)	1		
Yes	188 (47.59)	100 (55.25)	1.36 (1.09–1.69)		
Use of chloroquine or hydroxychloroquine				**0.115**	**0.014**
No	390 (98.73)	117 (97.79)	0.57 (0.36–0.89)		
Yes	5 (1.27)	4 (2.21)	1		
Use of Heparin (Enoxaparin)				0.165	0.074
No	68 (17.17)	24 (13.19)	1		
Yes	328 (82.83)	158 (86.81)	1.36 (0.97–1.92)		

CI: confidence interval; PR: adjusted prevalence ratio. The Wald test was used to calculate all “*p*” values, *p* < 0.05 was considered statistically significant (bold). Variables with *p* < 0.20 were later analyzed by multiple hierarchical Poisson regression.

**Table 2 vaccines-11-00071-t002:** Multiple regression analysis of the association of in-hospital death and sociodemographic data, chronic diseases, and treatment in patients with severe acute respiratory syndrome due to COVID-19 (*n* = 397).

	Hospital Death	
Variable	PR (CI 95%)	*p*
Gender		0.053
Male	1	
Female	0.80 (0.64–1.00)	
Age		0.001
20–59	1	
60 or more	1.55 (1.22–1.97)	
Diabetes mellitus		0.252
No	1	
Yes	1.13 (0.92–1.39)	
Pronation		0.237
No	1	
Yes	1.15 (0.91–1.45)	
Use of Azithromycin		0.012
No	1	
Yes	1.32 (1.06–1.64)	
Use of Heparin (Enoxaparin)		0.057
No	1	
Yes	1.40 (0,99–1.97)	

CI: confidence interval; PR: prevalence ratio. Wald’s test was used to calculate all *p*-values. *p* < 0.05 was considered statistically significant.

## Data Availability

Not applicable.

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
