# Peer review of "Mortality and Associated Factors in Patients with COVID-19: Cross-Sectional Study"

_vaccines, 2022, doi:10.3390/vaccines11010071_

Round 1
Reviewer 1 Report
The manuscript “Mortality and associated factors in patients with COVID-19: cross-sectional study could provide information about the prevalence of mortality and factors related to mortality due to COVID-19 in patients with severe acute respiratory syndrome (SARS) in a Central-West region of Brazil. It permits the comparison of these data with other worldwide regions to improve surveillance. The topic of this work is partially appropriated for Vaccines because several aspects related to vaccines were discussed here but vaccination is not adressed in the present study. I listed a number of comments which I feel should be improved.
Specific comments:
-L18, please check if the p=0.000 is appropriated.
-L18, considering that independent factors associated with mortality in COVID-19 patients were the age of >60 years and use of azithromycin. Is this appropriated to Vaccines MDPI?
-L18, could you provide if the independent risk related to >60 years and use of azithromycin is positive or negative?
-L24-31, other pathological factors than pneumonia are related to deaths in COVID-19. Could you provide it in the first paragraph?
-L24-L42, In my point of view the introduction needs to be improved with more information about the current knowledge about prevalence and factors related to mortality in patients wite severe acute respiratory syndrome (SARS) due to COVID-19 requiring IMV. In this context, vaccination is a critical topic and should be addressed here.
-L44, please, provide the city and state of the hospital.
-L57-68, in my point of view the analysis of the number of doses and/or time of vaccination should be analyzed here.
-L88, it is not clear why the variables of hyperthyroidism, mental illness, asthma, liver failure, smoking, and use of chloroquine/hydroxychloroquine were excluded.
-L96, the size of letter in figure 1 should be increased, in the first square the line is above the letters.
-L104, please improve the English grammar to “not having a diagnosis of”
-L105-108, in my point of view the protective factors described here should be inserted in the abstract.
-L110, some words as “sim “ and “não” are described in Portuguese and not in English in table 1.
-L109-110, the description of frequency and prevalence could be described better in legend of table 1.
-Table 1 , the value of zero is significative?
-The vaccination was described or analyzed here.
-Conclusion: in my point of view the protective factors described here should be inserted in the conclusion.
Author Response
-L18 Regarding p = 0.000, it was because we set it equal to the statistical program, however after its analysis we considered it to be quite valid and, therefore, we modified it to "p < 0.001".
-L18 We looked for other factors, which were not related to vaccination, but which were related to mortality. In addition, we consider the vaccination status of patients to be of paramount importance, but the medical records from which we obtained the data did not present information about immunizations for COVID-19, therefore, we maintained the scope of mortality and associated factors in these patients with acute respiratory syndrome serious by COVID-19.
-L18 Sorry, but I didn't understand what was asked.
-L24-31 I add more factors associated with in-hospital mortality from COVID19.
-L24-42 I add more information regarding risk factors and mortality from COVID-19, but the scope of this article does not address vaccination in its variables, so unfortunately I do not write more associations with vaccination due to the risk of losing the coherence of the ideas presented.
-L44 I add the city and state of the hospital.
- L57-68 I see no need for change, in order not to compromise the understanding of the results we found.
- L88 I see no need for change, in order not to compromise the understanding of the results we found.
-L96 Adjusted.
-L104 Ajusted.
-L 105-108 Ajusted.
- L 110 Ajusted.
Reviewer 2 Report
The article is very well written and well designed. The interpretation of the results and conclusions are clearly supported by the data provided. The reference list can be improved with a few relevant papers published on MDPI journals. In the Discussion, I invite the Authors to drop some lines on the comparison of data obtained with previous reports dealing with mortality of SARS-CoV-2 infection and the level of hospitalization in other territories (DOI: 10.3390/jcm11144196; DOI: 10.1136/bmj.m1835). Other interesting topic do be emphasized it is the excess of mortality related to under-reported deaths since many who died from COVID-19 were likely never tested or they had false negative RT-PCR results, especially during the first wave (DOI: 10.3390/healthcare9020119)
Author Response
I kindly add the indicated article to the discussion (DOI: 10.3390/jcm11144196; DOI: 10.1136/ bmj.m1835 ). In the second article, we did not identify a significant correlation with our topic, so we will keep it without adding it (DOI: 10.3390/ saúde9020119 ).
Round 2
Reviewer 1 Report
The author’s response is hard to understand. They did not insert the authors’responses together with reviewer comments and when the manuscript is corrected, the correction was not inserted with responses. This format is hard for reviewers and it is the first time that I received it in this format. This manuscript does not present significant advances for science, although apparently, it shows that data in a region of the Brazilian Midwest corroborate other parts of the country. The discussion does not seem so clear in this sense. Although the authors corrected identified flaws, the manuscript in general did not show a significant improvement. The question about the independent risk related to >60 years and the use of azithromycin was not improved. In the response L24-L42, the authors clearly demonstrate that they do not have data on vaccination, what is not understandable is why choose the journal Vaccines (ISSN 2076-393X). The novel sentence in the abstract is hard to read because the English grammar: please change “not having”,,, It is inadequate for sentences related to diseases.
In the novel version:
L61, the number 44 appears inappropriated.
L77, please change sex per gender.
L100, Unfortunately I hadn't noticed this error on the first reading, but in my point of view, the following sentence doesn't make sense: “Statistical significance was established using a cutoff value of p < 0.05.” . The cut-off can be the mean value plus one or two standard deviations, but it doesn't have a p-value.
L101-102- The correction of the sentence “the p-value of 0.001 indicates that the relationship is statistically significant at the α = 0.05 level.” If the alpha error is 0.05 the p-value of 0.001 is incorrect or needs to be clarified.
-L88 in the first version and 105-106 in present version, in my point of view the authors’ responses are unsatisfactory, it is not clear why the variables of hyperthyroidism, mental illness, asthma, liver failure, smoking, and use of chloroquine/hydroxychloroquine were excluded. It is important for the reader to understand the reason for the exclusion of these factors.
Author Response
Good afternoon,
Performed as suggested fixes. We identified a writing error in the section “In addition, the variables of hyperthyroidism, mental illness, asthma, liver failure, smoking and use of chloroquine/hydroxychloroquine were excluded”. As it does not make sense because some variables are part of our results.
Regarding the submission to the journal Vacinas, although the work does not portray the subject related to vaccination, the journal, after having seen the manuscript on the open access page, showed interest in the publication, which justifies the reason for our submission to it.
I hope the fixes meet your expectations. Thanks in advance for your consideration and corrections in order to improve the quality of our manuscript.
Sincerely.